# Vibration threshold in non-diabetic subjects

Svea Nolte[1]*, Marco van Londen[2], Jan Willem J. Elting[3], Bianca T. A. de Greef[4,5], Jan B. M. Kuks[3], Catharina G. Faber[4], Ilja M. Nolte[6], Rob J. M. Groen[1], Stephan J. L. Bakker[2], Dion Groothof[2], Ivonne Lesman-Leegte[1], Stefan P. Berger[2], Gea Drost[1,3]

**1** Department of Neurosurgery, University Medical Center Groningen and University of Groningen, Groningen, the Netherlands, **2** Division of Nephrology, Department of Internal Medicine, University Medical Center Groningen and University of Groningen, Groningen, the Netherlands, **3** Department of Neurology, University Medical Center Groningen and University of Groningen, Groningen, the Netherlands, **4** Department of Neurology, Maastricht University Medical Center and University of Maastricht, Maastricht, the Netherlands, **5** Klinische Epidemiologie en Medical Technology Assessment (KEMTA), Maastricht University Medical Center, Maastricht, the Netherlands, **6** Department of Epidemiology, University Medical Center Groningen and University of Groningen, Groningen, the Netherlands

* s.nolte@umcg.nl

**Data Availability Statement:** The anonymized datasets generated during and/or analyzed during the current study can be found in the supplements (S1 and S2 Datasets).

## Abstract

Measuring vibration perception threshold (VPT) accurately classifies and quantifies the severity of loss of vibration perception. A biothesiometer (Bio-thesiometer®; Bio Medical Instrument Co, Ohio, USA) appears to be the most suitable tool to determine VPT due to its low inter-rater variability and low occurence of adaption to the sensation. Different VPT values for a biothesiometer have been described, however, specification on age, height and different measurement locations is currently lacking. The objective of our study was to identify determinants of vibration perception in non-diabetic subjects, in order to provide individualized normal values of VPTs for clinical practice. Measurements of the vibration perception were performed on the big toes, insteps, lateral malleoli, and wrists. A total of 205 healthy subjects were included (108 (52.7%) males) with a median [interquartile range] age of 59 [51;64] (range 21–80) years. Mean height was 174.45 ± 9.20 cm and mean weight was 82.94 ± 14.84 kg, resulting in a mean BMI of 27.19 ± 4.00 kg/m². In stepwise forward linear regression analyses, age (st. β = 0.51, p < 0.001) and height (st. β = 0.43, p < 0.001) were found to be the independent unmodifiable determinants of the VPT at the big toe. Regression coefficients for quantiles of the determinants age and height were incorporated in the corresponding regression equations. This study provides equations to calculate age- and height-specific normal values for VPT that can be used in clinical practice and in large research studies.

## Introduction

Vibration sense is often impaired in large myelinated nerve fiber deficits such as in diabetic polyneuropathy [1, 2].

To examine vibration sense in a quantitative way vibration perception thresholds (VPTs) can be determined either by using a Rydel-Seiffer graduated tuning fork or a biothesiometer

**Funding:** The author(s) received no specific funding for this work.

**Competing interests:** The authors have declared that no competing interests exist.

[2–4]. When using a Rydel-Seiffer tuning fork the VPT can be measured with weights attached at 64 Hz or without at 128 Hz. The VPT determined with a Rydel-Seiffer tuning fork is defined as the value nearest to the point where vibration is no longer perceived. This is contrary to a biothesiometer which measures the VPT as the point at which the first sensation of the vibration appears [2, 4]. It has been shown for the upper limbs that by using the biothesiometer and thus detecting the first sensation of vibration, adaption to the sensation of vibration is less likely to develop [5]. Moreover, a recent study has shown that a handheld biothesiometer is a superior tool compared to the Rydel-Seiffer graduated tuning fork to monitor changes in vibration sense over time, which makes it ideal for long-term follow ups [5]. It is also noteworthy that a biothesiometer has a low inter-rater variability [6], particularly important in clinical practice and in large studies in which several investigators perform the measurements.

Different VPT values for a biothesiometer have been described [5, 7] and it appeared that in individuals without sensory disturbances VPT are not only influenced by age [5] but also by height [7]. Bloom et al. already mentioned that next to a strong correlation with age, height appeared to be of influence on VPT assessed at the feet [5]. Maffei et al. provided normal values depending on age and height but did not specify the location of the measurement [7]. As therapeutic approaches for large myelinted nerve fiber deficits are proceeding [6], specific normal values to follow up patients are crucial.

In this study, we aimed to provide individualized normal values for VPT in a group of non-diabetic subjects for different measurement locations on the body which can be used as reference values for clinical use.

## Materials and methods

### Participants

The inclusion of healthy subjects is displayed in Fig 1.

In this study, 260 subjects, during their screening for living kidney donation, were assessed for eligibility. Subjects were at least 18 years of age. We excluded subjects with vitamin B12 deficiency [8], alcohol intake of more than three units per day, missing blood values of vitamin B12, fasting plasma glucose or HbA1c, and patients with diabetes mellitus. Diabetes was defined as fasting plasma glucose higher or equal to 7.0 mmol/L, HbA1c above or equal to 48 mmol/mol, or the use of antidiabetic medication [9]. Other exclusion criteria were neurological disorders, trauma to limbs, and malignancies. However, these were not present in our subject population. All subjects underwent measurements in the University Medical Center Groningen (UMCG) as part of the TransplantLines Biobank and Cohort study, in which the living kidney donors serve as healthy controls [10]. We collected data on age, sex, height, weight, medical history, medication use [11–13], and co-morbidities from hospital records of the donor screening. Subjects were included between June 2016 and October 2017. All participants provided written informed consent. The study was approved by the local Medical Ethical Committee of the University Medical Center Groningen (METc 2014/077). All procedures are in accordance with the Declaration of Helsinki.

### Recording of vibration perception

To measure the vibration sensation in a group of non-diabetic subjects, we used a handheld biothesiometer (Bio-thesiometer®; Bio Medical Instrument Co, Ohio, USA) with a rubber applicator, which is the part that vibrates at a frequency of 120 Hz and with an amplitude range of 0.01–25.5 μm. The biothesiometer is calibrated regularly according to the manufacturers' recommendations. The vibration is not audible [4]. The measurements were performed in the morning by a group of 26 well-trained investigators. Before the measurements were

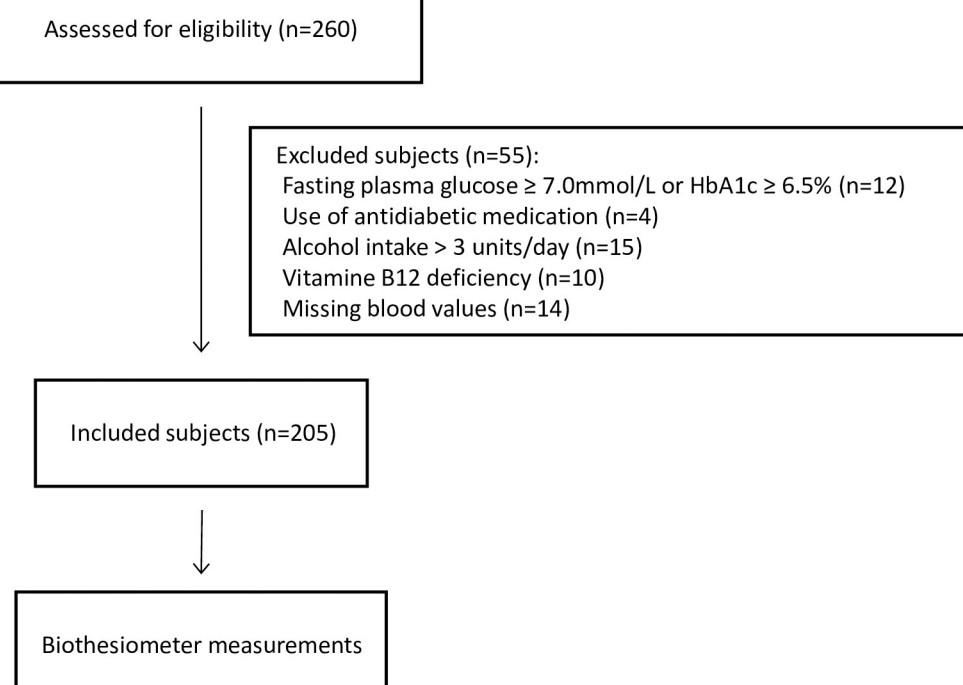

**Fig 1. Participants.**

performed, subjects were familiarized with the sensation of a biothesiometer by applying the vibration button to a hand, while increasing the amplitude of the vibrator button from zero to the highest amplitude possible. For the measurements, the subjects were lying on a bed after taking off their socks. During the measurements, the vibrator button was primarily applied to the tip of the big toes, and secondarily to three other points of the body, i.e. the insteps, lateral malleoli, and wrists (Fig 2). Subjects were asked to concentrate on feeling the vibration sensation and to report the first sensation of vibration by saying "Stop". Each so called 'threshold measurement' was tested bi-laterally two times (Fig 3). If the difference between the first two

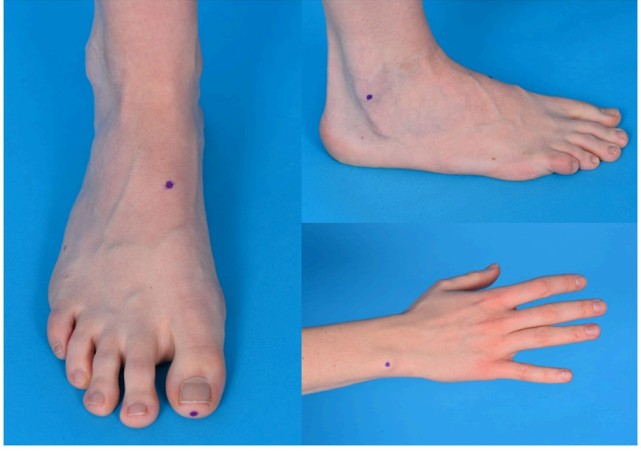

**Fig 2. Measured points (big toe, instep, lateral malleolus, wrist).**

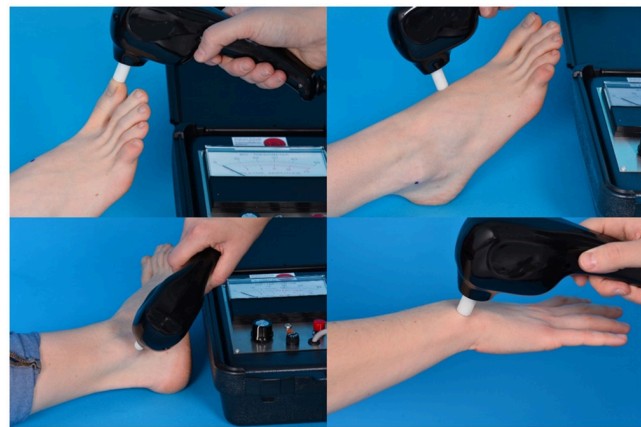

**Fig 3. Measurement.**

measurements at one side was more than 20%, the point was tested for a third time. If this was the case, the most deviating value was excluded. More detailed information on the number of subjects that underwent a third measurement is shown in Table 1. The mean value of all included measurements is defined as the threshold at the given point of measurement, used to determine the VPT. To translate the voltage readings obtained from the Biothesiometer scale into absolute amplitude values the calibration table of the manual was used. The VPT are given in microns of motion in which a micron equalizes $10^{-4}$ cm [4].

Additionally, in order to evaluate the inter-rater reliability of a biothesiometer, a group of 41 subjects underwent three repeated measurements, each performed by a different investigator. In total, 25 different investigators were involved in this part of the study.

## Blood samples

We collected data on fasting blood glucose, HbA1c, folic acid, and vitamin B12 concentrations. Analyses of fasting blood glucose and vitamin B12 were both done using a Roche Cobas 8000 platform (Roche, Mannheim, Germany). HbA1c levels were determined with a Tosoh G8 (ion exchange HPLC) (Tosoh/Sysmex Europe, Norderstedt, Germany).

## Statistical analysis

Descriptive statistics were used to characterize the study population. Continuous variables with a normal distribution were displayed as mean (standard deviation, SD). Continuous variables with a skewed distribution were displayed as median [interquartile range, IQR]. Categorical variables were displayed as frequencies with percentages. Distributions of continuous variables were checked with histograms, QQ plots, and Shapiro-Wilk test. As no clinical

**Table 1. Details on number of third measurement[1].**

| Big toes | | Insteps | | Lateral malleoli | | Wrists | |
|---|---|---|---|---|---|---|---|
| Left | Right | Left | Right | Left | Right | Left | Right |
| 32 (16%) | 26 (13%) | 47 (23%) | 29 (14%) | 35 (17%) | 34 (17%) | 20 (10%) | 13 (6%) |

[1]A third measurement was performed if the difference between the first two measurements at one side was >20%.

relevant side-to-side variation was found, it was decided to define the mean of all included measurements of both sides as threshold.

To identify clinical characteristics that function as unmodifiable determinants of the VPT, univariate linear regression and stepwise forward linear regression analyses were performed for the measurements at the big toes. The variables age, height, sex, weight, and smoking behaviour were tested. For the stepwise forward linear regression analyses the P-value to enter was defined as ≤ 0.05. To conduct linear regression analyses the obtained thresholds were log transformed because of non normal distributions.

By means of quantile regression models using the 95th percentile corresponding regression, equations were established to be able to calculate a normal VPT for each individual person dependent on all significant predictors. The dependent variable which corresponds to the VPT measured at the different locations was not log transformed because quantile regression is a non-parametrical test [14]. The 95th percentile corresponds to the 95% specificity. Regression equations were computed for all points of measurement: big toes, insteps, lateral malleoli, and wrists. Previous studies used quantile regressions to establish a corresponding regression equation [15, 16]. To assess the accuracy of the quantile regression model, the standard Koenker and Bassett method as well as bootstrap resampling with performing 2000 replications were conducted to provide standard errors [17]. When calculating the normal VPT values using the derived equations, the result will be given in microns of motion [4]. To determine the applicability of these equations the fifth percentile of age was used as a cut-off because of the few young subjects present in the study population. In the age group younger than 39 years only few subject per age were included and therefore the generalizability of these measurements seems not reliable. Consequently, these equations cannot be used for subjects younger than 39 years.

To explore the inter-rater reliability of a biothesiometer an intraclass correlation coefficient (ICC) was performed. ICC estimates and their 95% confident intervals were calculated based on a single rating, absolute agreement, and two-way random effects model ($ICC_{2,1}$). ICCs were interpreted along the following cut-offs: >0.9, excellent reliability; 0.75–0.9, good reliability; 0.5–0.75 moderate reliability; and <0.5, poor reliability [18].

A sample size of 119 subjects was needed to determine the 90% confidence interval of the 2.5th and 97.5th percentile of a population considering the use of non-parametric statistics [19].

SPSS version 23 for Windows (IBM, Armonk, USA), Stata 15 (StataCorp LCC, College Station, USA) and Graphpad Prism 7 for Windows (Graphpad, San Diego, CA), Python 3.0 (Python Software Foundation, Amsterdam, Netherlands) were used for statistical analyses. P-values below 0.05 were considered statistically significant.

## Results

### Participant characteristics

The mean and median of clinical characteristics of the 205 included subjects are summarized in Table 2. The subject age was 59 [51;64] years and 108 (52.7%) subjects were male. Their height and weight were 174.45 ± 9.20 cm and and 82.94 ± 14.84 kg respectively, resulting in a BMI of 27.19 ± 4.00 kg/m2. Of the subjects, 56 (27.3%) had hypertension, 15 (7.3%) had a history of cardiovascular disease, 112 (54.6%) used medication and 28 (13.7%) were current smokers. Alcohol intake was 0.47 [0;0.86] units per day. Blood glucose was 5.40 ± 0.54 mmol/L, HbA1c was 37 [35;39] mmol/mol (5.5 [5.4;5.7] %), vitamin B12 was 298.0 [234.0;371.5] pmol/L, and folic acid was 15.1 [11.4;22.0] nmol/L.

**Table 2. Clinical characteristics of the cohort.**

| Variable[1] | | Range |
|---|---|---|
| Number | 205 | |
| Age, years | 59 [51;64] | 21–80 |
| Sex, n(%) male | 108 (52.7) | |
| Height, cm | 174.45 ± 9.20 | 152.50–195.50 |
| Weight, kg | 82.94 ± 14.84 | 54.00–123.80 |
| BMI, kg/m$^2$ | 27.19 ± 4.00 | 18.90–40.90 |
| Systolic blood pressure, mmHg | 127.32 ± 15.05 | 93.00–175.00 |
| Diastolic blood pressure, mmHg | 74.90 ± 10.93 | 49.00–100.00 |
| Hypertension, n(%) | 56 (27.3) | |
| History of cardiovascular disease, n(%) | 15 (7.3) | |
| Medication use, n(%) | 112 (54.6) | |
| Statins, n(%) | 20 (9.8) | |
| Antiretroviral medication, n(%) | 0 (0) | |
| Smoking, n(%) | 28 (13.7) | |
| Alcohol intake, units/day | 0.47 [0;0.86] | 0–2.86 |
| *Relevant blood values* | | |
| Blood glucose, mmol/L | 5.40 ± 0.54 | 3.70–6.80 |
| HbA1c, mmol/mol | 37 [35;39] | 30–47 |
| Vitamin B12, pmol/L | 298.0 [234.0;371.5] | 146.0–750.0 |
| Folic acid, nmol/L | 15.1 [11.4;22.0] | 3.2–45.4 |

[1]Continuous variables with a normal distribution are given as mean ± standard deviation; continuous variables wih a skewed distribution are given as median [interquartile range]; categorical variables are given as frequencies (percentage).

## Determinants of the VPT

The results of the univariate linear regression analyses are summarized in Table 3.

The results of the analyses of determinants of VPT at the big toe are summarized in Table 4. In stepwise forward linear regression analyses, age (st. $\beta$ = 0.51, p < 0.001) and height (st. $\beta$ = 0.43, p < 0.001) were significantly associated with VPT at the big toe, $F(2,202) = 52.724$, p < 0.001, $R^2 = 0.343$. As independent determinants, age and height were as unmodifiable factors included in the regression equations to calculate the normal values of the VPT.

The results of the linear regression analyses at the other points of measurement are shown in Table 4.

## VPT

Figs 4 and 5 show the relationship between the VPT and the independent unmodifiable determinants of VPT at the big toes, i.e. age and height. The distribution of subjects dependent on age and height and the measured VPT is displayed in Table 5.

Table 6 provides the unstandardized regression coefficients of the slope for the variables age and height obtained in the multivariable quantile regression model. This model estimates the normal value of the VPT as a function of age and height, with which we can write the

**Table 3. Univariate linear regression analyses of determinants of VPT.**

| Measurement location | Independent variable | B (95% CI) | Standardized $\beta$ | P-value |
|---|---|---|---|---|
| **Big toe** | Age, years | 0.02 (0.02–0.03) | 0.47 | <0.001 |
| | Height, cm | 0.02 (0.01–0.03) | 0.32 | <0.001 |
| | Weight, kg | 0.01 (0.01–0.02) | 0.33 | <0.001 |
| | Sex | -0.29 (-0.42- -0.15) | -0.28 | <0.001 |
| | Smoking behavior | 0.03 (-0.18–0.24) | 0.02 | 0.77 |
| **Instep** | Age, years | 0.03 (0.02–0.03) | 0.48 | <0.001 |
| | Height, cm | 0.02 (0.01–0.02) | 0.27 | <0.001 |
| | Weight, kg | 0.01 (0.01–0.02) | 0.33 | <0.001 |
| | Sex | -0.33 (-0.47- -0.18) | -0.30 | <0.001 |
| | Smoking behavior | 0.07 (-0.15–0.29) | 0.05 | 0.51 |
| **Lateral malleolus** | Age, years | 0.02 (0.02–0.03) | 0.48 | <0.001 |
| | Height, cm | 0.01 (0.01–0.02) | 0.27 | <0.001 |
| | Weight, kg | 0.01 (0.01–0.02) | 0.34 | <0.001 |
| | Sex | -0.31 (-0.44- -0.18) | -0.32 | <0.001 |
| | Smoking behavior | 0.01 (-0.19–0.21) | 0.01 | 0.92 |
| **Wrist** | Age, years | 0.01 (0.01–0.02) | 0.36 | <0.001 |
| | Height, cm | 0.004 (-0.002–0.01) | 0.09 | 0.21 |
| | Weight, kg | 0.004 (0.001–0.01) | 0.16 | 0.02 |
| | Sex | -0.18 (-0.27- -0.08) | -0.24 | 0.001 |
| | Smoking behavior | 0.07 (-0.08–0.22) | 0.07 | 0.36 |

CI: confidence interval.

corresponding regression equations:

$$\Upsilon_{\text{VPT\_BIGTOE}} = -90.09 + 0.24 \times \text{AGE} + 0.50 \times \text{HEIGHT}$$

$$\Upsilon_{\text{VPT\_INSTEP}} = -98.70 + 0.35 \times \text{AGE} + 0.52 \times \text{HEIGHT}$$

**Table 4. Forward selection multiple linear regression models of determinants of VPT[1].**

| Measurement location | Independent variable | B (95% CI) | Standardized $\beta$ | P-value |
|---|---|---|---|---|
| **Big toe** | Age, years | 0.03 (0.02–0.03) | 0.51 | <0.001 |
| | Height, cm | 0.03 (0.02–0.03) | 0.43 | <0.001 |
| **Instep** | Age, years | 0.03 (0.02–0.03) | 0.50 | <0.001 |
| | Height, cm | 0.02 (0.01–0.03) | 0.30 | <0.001 |
| | Weight, kg | 0.01 (0–0.01) | 0.17 | 0.02 |
| **Lateral malleolus** | Age, years | 0.02 (0.02–0.03) | 0.50 | <0.001 |
| | Height, cm | 0.02 (0.01–0.02) | 0.25 | 0.001 |
| | Weight, kg | 0.01 (0–0.01) | 0.19 | 0.01 |
| **Wrist** | Age, years | 0.01 (0.01–0.02) | 0.30 | <0.001 |
| | Sex | -0.18 (-0.29- -0.08) | -0.23 | 0.001 |

[1] Forward selection, P-value to enter ≤ 0.05.
CI: confidence interval.

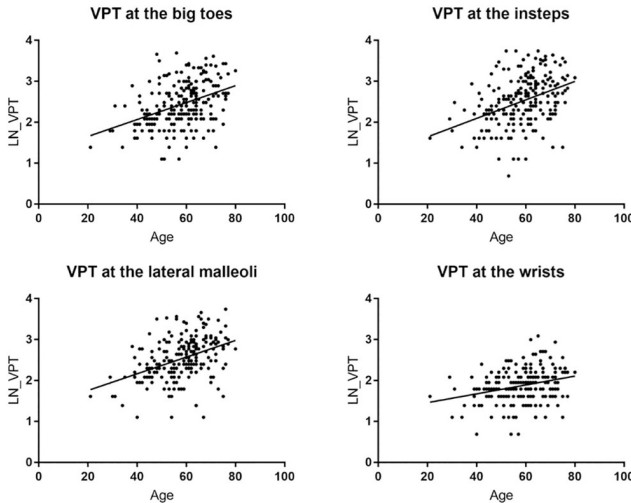

**Fig 4. VPT against age for big toes, insteps, lateral malleoli, and wrists.**

$$\Upsilon_{\text{VPT\_LATMALLEOLUS}} = -70.27 + 0.26 \times \text{AGE} + 0.37 \times \text{HEIGHT}$$

$$\Upsilon_{\text{VPT\_WRIST}} = -4.55 + 0.06 \times \text{AGE} + 0.02 \times \text{HEIGHT}$$

The results of the bootstrapped model are summarized in Table 6. Figs 6 and 7 show conditional quantiles of VPT as a function for age and height for the different measurement locations.

In clinical practice the provided nomograms can be used: by filling in age and height of the patient on the outer scales, connecting the two points by drawing a straight line, the normal value for the VPT can be read from the middle scale (S1 Fig).

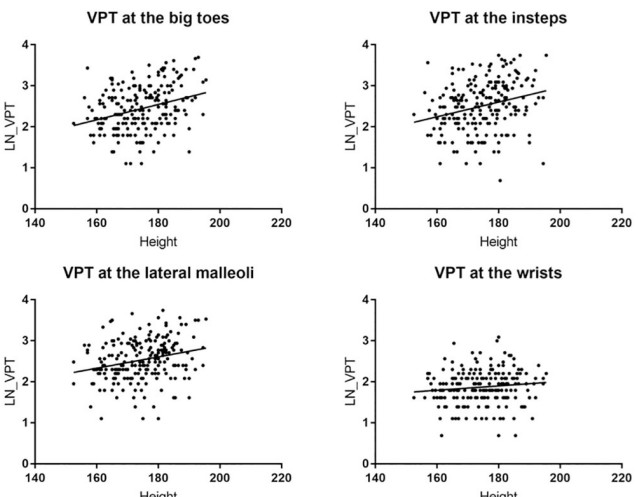

**Fig 5. VPT against height for big toes, insteps, lateral malleoli, and wrists.**

**Table 5. Distribution of subjects an measured VPT dependent on age and height grouping.**

| Age, years | Number of subjects | Big toes[1] | Insteps[1] | Lateral malleoli[1] | Wrists[1] |
|---|---|---|---|---|---|
| ≤ 39 | 10 | 0.79 (0.10–1.48) | 0.94 (0.19–1.69) | 0.83 (0.19–1.44) | 0.30 (0.15–0.46) |
| 40–49 | 39 | 1.70 (0.73–2.67) | 1.51 (0.81–2.22) | 1.52 (0.91–2.13) | 0.41 (0.34–0.49) |
| 50–59 | 59 | 1.92 (1.28–2.58) | 2.66 (1.68–3.65) | 2.36 (1.62–3.10) | 0.48 (0.39–0.57) |
| 60–69 | 69 | 2.94 (2.22–3.66) | 3.94 (2.89–5.00) | 3.50 (2.54–4.47) | 0.75 (0.55–0.95) |
| 70–80 | 28 | 3.67 (2.24–5.10) | 4.45 (2.75–6.15) | 3.73 (2.43–5.04) | 0.80 (0.52–1.07) |
| <160 | 10 | 1.64 (-0.45–3.73) | 2.11 (-0.53–4.74) | 1.41 (0.68–2.13) | 0.49 (0.33–0.66) |
| 160–169 | 56 | 1.57 (1.14–1.99) | 2.07 (1.27–2.87) | 2.17 (1.27–3.07) | 0.52 (0.37–0.67) |
| 170–179 | 74 | 1.85 (1.39–2.32) | 2.48 (1.82–3.15) | 2.22 (1.70–2.75) | 0.61 (0.47–0.75) |
| 180–190 | 55 | 3.51 (2.53–4.50) | 4.67 (3.33–6.01) | 3.72 (2.67–4.77) | 0.67 (0.47–0.87) |
| >190 | 10 | 5.74 (1.40–10.08) | 4.31 (0.33–8.28) | 4.68 (1.23–8.13) | 0.52 (0.28–0.76) |

[1] mean VPT (95% CI).

## Inter-rater reliability

ICC revealed an excellent reliability of a biothesiometer measurement at the big toe (ICC = 0.92, 95% CI = 0.87–0.95) and the instep (ICC = 0.90, 95% CI = 0.85–0.94), and a good reliability at the lateral malleoli (ICC = 0.85, 95% CI = 0.76–0.91) and the wrist (ICC = 0.82, 95% CI = 0.72–0.89).

## Discussion

In this study, we were able to provide individualized normal values for VPT using a biothesiometer which was shown to have a good to excellent inter-rater reliability. Moreover, this study showed that VPTs are influenced by age and height which were included when determining the normal values.

We found that with increasing age the vibration perception decreases at all points of measurement. Age-dependency of vibration perception has been first reported by Pearson et al.

**Table 6. Quantile regression.**

| | Variable | Original model[1] | | | | Model after bootstrap resampling[2] | | | |
|---|---|---|---|---|---|---|---|---|---|
| | | RC | SE | 95% CI | P-value | RC | SE | 95% CI | P-value |
| **Big toe** | Constant | -90.09 | 34.35 | -157.82- -22.36 | 0.009 | -90.09 | 15.78 | -121.21- -58.98 | 0.006 |
| | Age, years | 0.24 | 0.16 | -0.07–0.54 | 0.132 | 0.24 | 0.09 | 0.07–0.40 | <0.001 |
| | Height, cm | 0.50 | 0.18 | 0.15–0.85 | 0.006 | 0.50 | 0.11 | 0.28–0.71 | <0.001 |
| **Instep** | Constant | -98.70 | 35.76 | -169.21- -28.19 | 0.006 | -98.70 | 23.23 | -144.51- -52.89 | <0.001 |
| | Age, years | 0.35 | 0.16 | 0.03–0.67 | 0.030 | 0.35 | 0.09 | 0.17–0.54 | <0.001 |
| | Height, cm | 0.52 | 0.19 | 0.16–0.89 | 0.005 | 0.52 | 0.14 | 0.24–0.81 | <0.001 |
| **Lateral malleolus** | Constant | -70.27 | 34.94 | -139.17- -1.37 | 0.046 | -70.27 | 13.06 | -96.03- -44.52 | <0.001 |
| | Age, years | 0.26 | 0.16 | -0.05–0.57 | 0.100 | 0.26 | 0.07 | 0.13–0.40 | <0.001 |
| | Height, cm | 0.37 | 0.18 | 0.02–0.73 | 0.041 | 0.37 | 0.08 | 0.21–0.54 | <0.001 |
| **Wrist** | Constant | -4.55 | 9.45 | -23.19–14.09 | 0.631 | -4.55 | 5.36 | -15.11–6.02 | 0.397 |
| | Age, years | 0.06 | 0.04 | -0.03–0.14 | 0.164 | 0.06 | 0.02 | 0.02–0.10 | 0.004 |
| | Height, cm | 0.02 | 0.05 | -0.08–0.12 | 0.703 | 0.02 | 0.03 | -0.04–0.08 | 0.525 |

[1] Standard Koenker and Bassett method;

[2] Bootstrap resampling (number of replications: 2000); RC: regression coefficient; SE: standard error; CI: confidence interval.

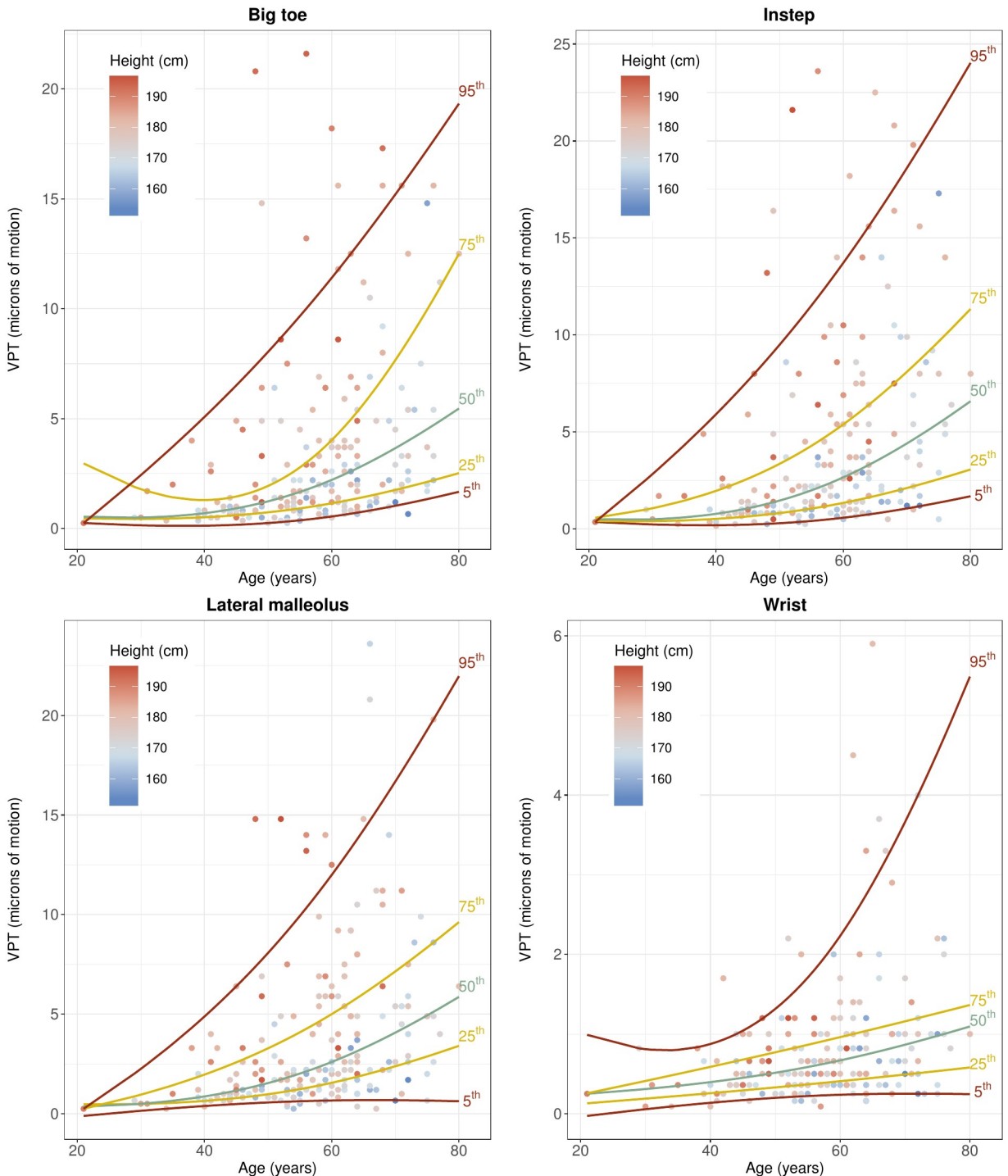

**Fig 6. Conditional quantiles of VPT as a function of age for different measurement locations.** Conditional quantiles were modelled flexibly using natural cubic splines of age with two degrees of freedom. Natural boundary conditions were imposed on the 2.5% and 97.5% quantiles of age. The specific quantiles associated with either of the lines is indicated on the right-hand edge of the line. The height across the participants is depicted as a color gradient, with lower values of height given in blue and higher values in red.

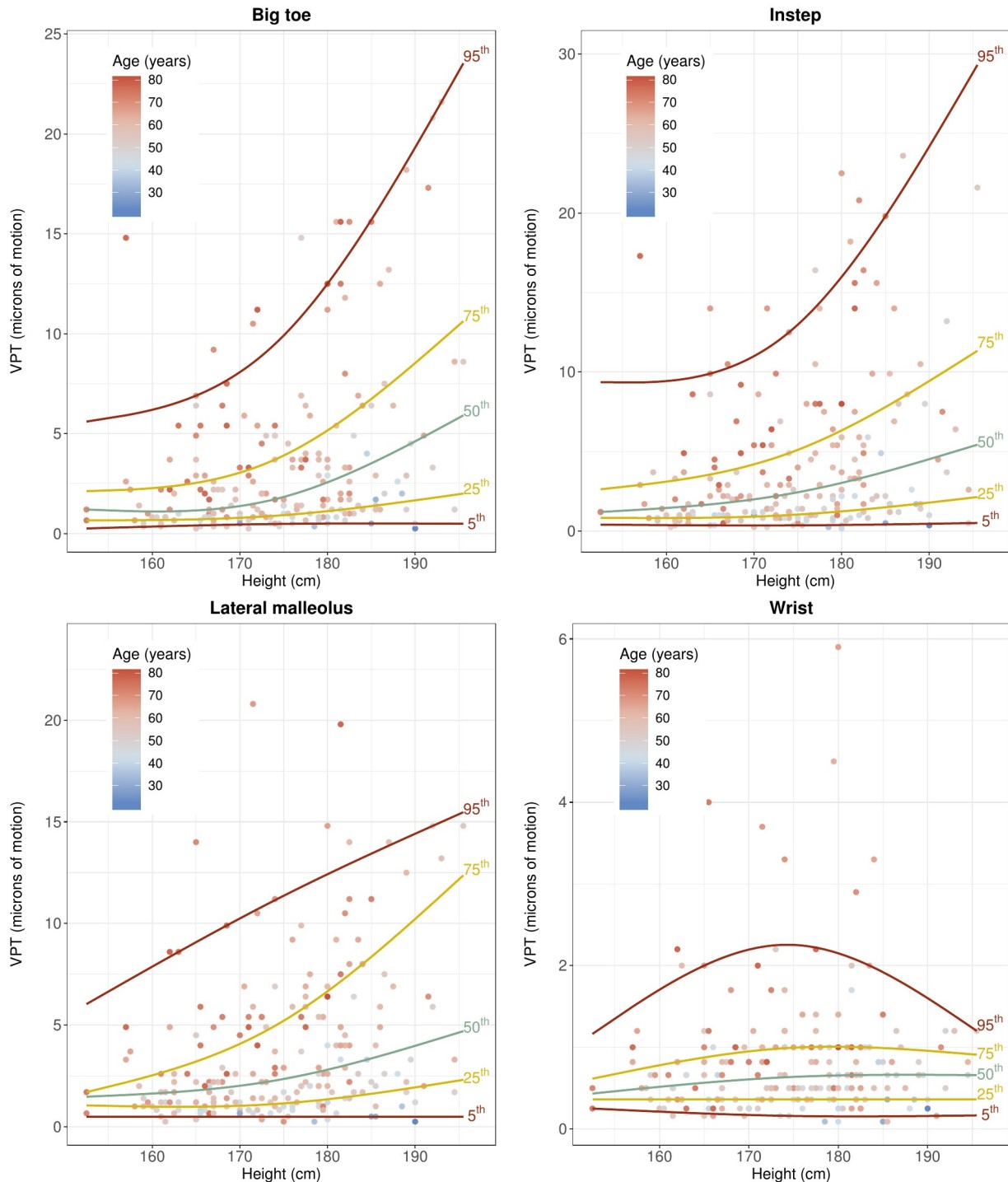

**Fig 7. Conditional quantiles of VPT as a function of height for different measurement locations.** Conditional quantiles were modelled flexibly using natural cubic splines of height with two degrees of freedom. The specific quantiles associated with either of the lines is indicated on the right-hand edge of the line. The age across the participants is depicted as a color gradient, with lower values of age given in blue and higher values in red.

[20] and has been confirmed by several other studies [2, 4–7, 21–24]. However, it is not clear what causes this reduction of vibration perception [2]. Vibration is detected by tactile receptors that are located in the epidermis and dermis [25]. When focusing on the detection of vibration, two different receptor systems are known: Pacinian corpuscles and Meissner's corpuscles [25–27]. Both deteriorate and are less present in ageing skin [25, 27]. Furthermore, it is known that with ageing myelin alterations take place in peripheral nerves: segmental de- and remyelination, axonal degeneration and regeneration are seen more often. This is possibly due to atherosclerotic vascular changes that happen during aging [28]. Besides atherosclerotic vascular changes, also a generalized reduction in vasculature of the dermis is observed in the elderly [26]. There are also remarkable changes in the central nervous system such as cerebral small vessel disease which are linked to mobility impairment in the elderly [29]. These vascular changes in the central and peripheral nervous system could also influence the conduction of vibration stimuli. Both, Pacinian corpuscles and Meissner's corpuscles, are supplied by Aβ nerve fibers [27, 30]. Demyelination or a decrease in thickness of the myelin possibly causes abnormal Aß-fiber conduction velocities [30]. The above described changes are a possible explanation for the age-dependency of vibration perception. Interestingly, two-point discrimination, another component of sensory testing examining the nerve innervation density, has been shown to be correlated to vibration sense and also deteriorates with ongoing age [31].

Height was also an independent determinant: with increasing height the vibration perception decreases at all points of measurement. This height relation can also be observed at the different points of measurement. VPTs at the wrist are lower than at the big toe, instep, or lateral malleolus, which might be linked to a shorter distance between wrist and spinal cord. So far, only Deshpande et al. found height to be an independent determinant of vibration perception [24]. Noteworthy is that according to a recent study, Dutch men are tallest of the world and Dutch women are second tallest after Latvian women [32]. As our study was carried out in the Netherlands, the variation of height within the subject population is likely to be larger than in previously performed studies in other countries. This may add to the association with height that we found.

Strengths of this study include the detailed database in which blood values, medication history and comorbidities are recorded for all subjects. Moreover, the measurements were performed under standardized conditions meaning always in the same environment and location. This is crucial because by applying standardized conditions and providing good training of the different investigators, it is likely that factors that could have influenced the vibration energy, as difference in weight through applying pressure when measuring with a biothesiometer, were ruled out. Moreover, subjects suffering from peripheral neuropathy caused by vitamin B12 deficiency, alcohol abuse or diabetes mellitus were excluded. Furthermore, it was shown that the inter-rater variability of a biothesiometer is low, making it an ideal device for daily clinical practice and large research studies [5]. Nevertheless, it should be noticed that quantitative sensory testing including testing of vibratory stimuli as an objective physical event requires a response of the subject which is a subjective report. As a consequence, there are limitations inherent to the method due to reaction time of the subject and the ability to focus on the task [33]. One of the weaknesses of our study is the low sample size for the group younger than 39 years. Consequently, it was not possible to establish regression equations that are applicable for this group. When determining vibration perception in clinical practice in a patient younger than 39 years, it is possible to use the normal values for a patient of 39 years. If this patient scores abnormal according to the used normal value, an abnormality in vibration perception is certain. However, a determined normal vibration perception would not be evident. Nevertheless, it should be noticed that peripheral neuropathies, which are among other sensory disturbances characterized by loss of vibration sense, have an increasing incidence from the age of

40 years onwards. The highest incidence was reported in the age group of 75 to 79 years [34]. Therefore, normal values for VPT which is one part of sensory testing when diagnosing peripheral neuropathies are in particular needed for the elderly. Another weakness of our study is that skin temperature at the different sites of measurement, which possibly affects the vibration perception, was not taken into account [21, 22].

Clinical applications of VPT are diseases involving the sensory nerves and especially those primarily affecting large myelinated fibers. Polyneuropathy is one of the most common afflictions. Considering the broad prevalence of polyneuropathy in the population, the continued trend in population ageing, and the increase of incidence with age, age specific thresholds are specifically important [34]. Moreover, it has been reported that presence of polyneuropathy is associated with a significant reduction in quality of life [35]. As polyneuropathy has such a great impact on the life of patients, it is important to examine the affliction in clinical practice by capturing symptoms. Loss of vibration perception can be detected using the regression equations for a biothesiometer that were established in the current subject population. When testing a patient's vibration perception and there is no perception at the big toes, the different equations enable the testing of more proximal locations. The biothesiometer can be applied in daily clinical practice for the quantification of the loss of vibration perception and in large research studies. Larger studies in a younger subject population are needed to extend the results found in this study thereby improving the reliability of the different VPTs. Furthermore, future prospective studies need to validate these regression equations in different healthy subject groups to establish a precise equation for clinical use. The next step is to test these equations in different patient populations, such as one with diseases like diabetes mellitus, one using neurotoxic medication, or one with abnormal nerve conduction studies.

In conclusion, this study provides normal values for VPT based on unmodifiable determinants, age and height. The established equations can be used to determine loss of vibration perception in clinical practice.

## Supporting information

**S1 Fig. Nomograms.** By filling in age and height of the patient on the outer scales, connecting the two points by drawing a straight line, the normal value for the VPT (vibration perception threshold) can be read from the middle scale.
(PDF)

**S1 File. Sensitivity analysis.** Patients who underwent a third measurement were excluded from the study population and the statistical analyses were carried out as described in the Methodology section.
(DOCX)

**S1 Dataset. The first dataset was used to determine unmodifiable determinants of VPT.**
(SAV)

**S2 Dataset. The second dataset was used to explore the inter-rater reliability of a biothesiometer.**
(SAV)

## Acknowledgments

This study was performed using the infrastructure and data provided by the TransplantLines Biobank and Cohort Study, which is registered at ClinicalTrials.gov under Identifier NCT03272841. We thank dr. R. Bakels, neurophysiologist at the UMCG, for advice regarding

the content of the manuscript. Moreover, we thank Johan Schneiders for creating the nomograms and Piet H. Toonder, medical photographer at the UMCG, for the visualization of a biothesiometer and the measurements.

## Author Contributions

**Conceptualization:** Svea Nolte, Marco van Londen, Ivonne Lesman-Leegte, Gea Drost.

**Formal analysis:** Svea Nolte, Jan Willem J. Elting, Bianca T. A. de Greef.

**Investigation:** Svea Nolte, Marco van Londen.

**Methodology:** Svea Nolte, Marco van Londen, Jan Willem J. Elting, Bianca T. A. de Greef, Ilja M. Nolte, Stephan J. L. Bakker, Gea Drost.

**Project administration:** Stephan J. L. Bakker.

**Supervision:** Marco van Londen, Ivonne Lesman-Leegte, Gea Drost.

**Visualization:** Dion Groothof.

**Writing – original draft:** Svea Nolte.

**Writing – review & editing:** Marco van Londen, Jan Willem J. Elting, Bianca T. A. de Greef, Jan B. M. Kuks, Catharina G. Faber, Rob J. M. Groen, Stephan J. L. Bakker, Ivonne Lesman-Leegte, Stefan P. Berger, Gea Drost.

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
