## [Decision Letter · Decision Letter 0]

7 May 2020

PONE-D-20-09300

Vibration thresholds in non-diabetic subjects

PLOS ONE

Dear Mrs. Nolte,

Thank you for submitting your manuscript to PLOS ONE. After careful consideration, we feel that it has merit but does not fully meet PLOS ONE’s publication criteria as it currently stands. Therefore, we invite you to submit a revised version of the manuscript that addresses the points raised during the review process.

As this manuscript was a resubmission of a previous one, we have invited to the reviewers of the old manuscript to provide a revision of the new manuscript. Their reports are very positive, both considering that the manuscript has highly improved. Additionally, two new reviewers, unaware of the previous submission, were also invited to review the manuscript. Their comments are also rather positive, but some remarks are still to be addressed. In general, most of the comments imply minor changes in order to make it clearer for the readers and better situate your study in the general field of tactile thresholds in this population.

We would appreciate receiving your revised manuscript by Jun 21 2020 11:59PM. To enhance the reproducibility of your results, we recommend that if applicable you deposit your laboratory protocols in protocols.io, where a protocol can be assigned its own identifier (DOI) such that it can be cited independently in the future. For instructions see: http://journals.plos.org/plosone/s/submission-guidelines#loc-laboratory-protocols

We look forward to receiving your revised manuscript.

Kind regards,

Inmaculada Riquelme

Academic Editor

PLOS ONE

Journal Requirements:

Reviewers' comments:

Reviewer's Responses to Questions

**Comments to the Author**

1. Is the manuscript technically sound, and do the data support the conclusions?

Reviewer #1: Yes

Reviewer #2: Yes

Reviewer #3: Yes

Reviewer #4: Partly

2. Has the statistical analysis been performed appropriately and rigorously? 

Reviewer #1: Yes

Reviewer #2: Yes

Reviewer #3: Yes

Reviewer #4: Yes

3. Have the authors made all data underlying the findings in their manuscript fully available?

Reviewer #1: Yes

Reviewer #2: No

Reviewer #3: No

Reviewer #4: Yes

4. Is the manuscript presented in an intelligible fashion and written in standard English?

Reviewer #1: Yes

Reviewer #2: Yes

Reviewer #3: Yes

Reviewer #4: Yes

5. Review Comments to the Author

Reviewer #1: This is a hugely interesting study. Patient population is clear and so are methods and results.

Minor comments:

1. Please shorten the Introduction.

This is a hugely interesting study. Patient population is clear and so are methods and results.

Minor comments:

1. Please shorten the Introduction.

This is a hugely interesting study. Patient population is clear and so are methods and results.

Minor comments:

1. Please shorten the Introduction.

Reviewer #2: Thank you for the invitation to review this paper with the title “Vibration thresholds in non-diabetic subjects.”

The paper presents how vibration perception thresholds are effected by age and height on. The

The Experiments, statistics, and other analyses are done in a proper way and are described in detail.

In general the conclusions are supported by the data presented.

I have only some minor questions, comments, and suggestions:

1. Data on tables 3-6 could be presented in one table. There are a lot of small tables also in the supplement part that could be presented together.

2. Are the regression equations presented for logarithmic values? The authors state that “the obtained thresholds were log transformed”

3. Regarding last line in conclusion: (“The established equations can be used to determine loss of vibration perception in clinical practice.”) Is it possible to calculate standard deviations from these equations? That would be needed if one would use them to determine a specific threshold that is above normal. (Figures 6-7) could be used for that purpose.

4. Line 57 “By using the biothesiometer and thus detecting the first sensation of vibration, adaption to the sensation of vibration is less likely to develop” According to reference 5, this is true for upper limbs but I can not see that this was the case for measurements in the lower limb.

Reviewer #3: Please make note of the published study by Rinkel et al. on normative data, with decline in sensibility, of static, moving and one-point discrimination.

Please make a statement on comparability or superiority of VPT measurements measured with the Biothesiometer compared to Rydel Seiffer tuning fork.

See further comments in the attached document.

Reviewer #4: These are my comments. i hope you find them useful

1. Sample size is too small. How did you come up with this sample size?

2. Value sof VPT are equipment specific and machines are not standardized. It sa common observation that the values form 2 different machines are not the same. Can the resuls be generalized to other equipments

3. Mean value of VP is less than 2, whereas for diabetic neuropathy thresholds such as 15 and 25 are used. Given the low mean value, minor differences after adustment for age and hieght are practically non-significant and unikely to be benificial in any clinical condition. In addition VPT values are partly subjective with significant variation and minor differences are unlikely to be of any importance. Which cliincal condition do you think your research is going to be beneficial?

4. Median age of the study population of 59 years is somewhat strange to generate normative values and adjustments for age/height. You shoud have chosen a healthy populaion below 50 yars of age

5. VPT is not as sensitive a test for diabetic neuroathy compared to nerve conduction studies. Can you enumeate which clinical condions is VPT preferred as diagnostic test for early detection

6. There are too many figures and tables which need to be cut-down to 1-2 each

6. PLOS authors have the option to publish the peer review history of their article (what does this mean?). If published, this will include your full peer review and any attached files.

Reviewer #1: No

Reviewer #2: No

Reviewer #3: Yes: Willem D. Rinkel MD

Reviewer #4: Yes: Shivaprasad Channabasappa

---

## [Author Response · Author response to Decision Letter 0]

19 Jun 2020

In response to the comments of the editor, we changed the file naming according to PLOS ONE's style requirements and added the data sets used to reach the conclusions of our manuscript to the Supplement Information. Please find attached a detailed, itemized, point-by-point response to the suggestions and concerns raised by the reviewers. In response to the comment of reviewer #1, we shortened the introduction which led to a more coherent structure. In response to the comments of reviewer #2, we added more clarifying information on the different statistical methods used to establish the regression equations for the normal values. To accommodate the comments of reviewer #3, we elaborated on the advantages of using a biothesiometer to measure vibration sense. Furthermore, we added more specific information on the study design including the measuring of vibration perception, the clinical parameters included in the prediction model, and the inter-rater reliability. In response to the comments of reviewer #4, we explained our decision for the given sample size and to what extent the high median age of the included population is problematic in clinical practice. Furthermore, we described the clinical implications of the provided normal values for a biothesiometer.

---

## [Decision Letter · Decision Letter 1]

15 Jul 2020

PONE-D-20-09300R1

Vibration thresholds in non-diabetic subjects

PLOS ONE

Dear Dr. Nolte,

Thank you for submitting your manuscript to PLOS ONE. After careful consideration, we feel that it has merit but does not fully meet PLOS ONE’s publication criteria as it currently stands. Therefore, we invite you to submit a revised version of the manuscript that addresses the minor points raised during the review process.

We look forward to receiving your revised manuscript.

Kind regards,

Inmaculada Riquelme

Academic Editor

PLOS ONE

Reviewers' comments:

Reviewer's Responses to Questions

**Comments to the Author**

1. If the authors have adequately addressed your comments raised in a previous round of review and you feel that this manuscript is now acceptable for publication, you may indicate that here to bypass the “Comments to the Author” section, enter your conflict of interest statement in the “Confidential to Editor” section, and submit your "Accept" recommendation.

Reviewer #1: All comments have been addressed

Reviewer #3: All comments have been addressed

2. Is the manuscript technically sound, and do the data support the conclusions?

Reviewer #1: Yes

Reviewer #3: Yes

3. Has the statistical analysis been performed appropriately and rigorously? 

Reviewer #1: Yes

Reviewer #3: Yes

4. Have the authors made all data underlying the findings in their manuscript fully available?

Reviewer #1: Yes

Reviewer #3: Yes

5. Is the manuscript presented in an intelligible fashion and written in standard English?

Reviewer #1: Yes

Reviewer #3: Yes

6. Review Comments to the Author

Reviewer #1: no further comments no further comments no further comments no further comments no further comments no further comments

Reviewer #3: It is wise to state that the Rydel Seiffer tuning fork with weights attached tests at 64 hz. Without weights, one tests at 128 Hz.

7. PLOS authors have the option to publish the peer review history of their article (what does this mean?). If published, this will include your full peer review and any attached files.

Reviewer #1: No

Reviewer #3: No

---

## [Author Response · Author response to Decision Letter 1]

29 Jul 2020

Reviewer #3

Comment 1:

It is wise to state that the Rydel-Seiffer tuning fork with weights attached tests at 64 Hz. Without weights, one tests at 128 Hz.

Response to comment 1:

We thank the reviewer for this comment. Accordingly, we added the statement on the Rydel-Seiffer tuning fork to the Introduction section in which we compared the testing procedure of a Rydel-Seiffer tuning fork to the one of a biothesiometer (see manuscript lines 51-52).

---

## [Editor Report · Decision Letter 2]

3 Aug 2020

Vibration thresholds in non-diabetic subjects

PONE-D-20-09300R2

Dear Dr. Nolte,

We’re pleased to inform you that your manuscript has been judged scientifically suitable for publication and will be formally accepted for publication once it meets all outstanding technical requirements.

Kind regards,

Inmaculada Riquelme

Academic Editor

PLOS ONE
---

## [Editor Report · Acceptance letter]

26 Aug 2020

PONE-D-20-09300R2 

Vibration threshold in non-diabetic subjects 

Dear Dr. Nolte:

I'm pleased to inform you that your manuscript has been deemed suitable for publication in PLOS ONE. Congratulations! Your manuscript is now with our production department. 

Kind regards, 

on behalf of

Dr. Inmaculada Riquelme 

Academic Editor

PLOS ONE